# A spatially uniform illumination source for widefield multi-spectral optical microscopy

**İris Çelebi**[1][◉], **Mete Aslan**[1][◉], **M. Selim Ünlü**[1,2,3]*

**1** Department of Electrical and Computer Engineering Boston University, Boston, MA, United States of America, **2** Department of Biomedical Engineering, Boston University, Boston, MA, United States of America, **3** iRiS Kinetics Inc, Boston, MA, United States of America

◉ These authors contributed equally to this work.
* selim@bu.edu

## Abstract

Illumination uniformity is a critical parameter for excitation and data extraction quality in widefield biological imaging applications. However, typical imaging systems suffer from spatial and spectral non-uniformity due to non-ideal optical elements, thus require complex solutions for illumination corrections. We present Effective Uniform Color-Light Integration Device (EUCLID), a simple and cost-effective illumination source for uniformity corrections. EUCLID employs a diffuse-reflective, adjustable hollow cavity that allows for uniform mixing of light from discrete light sources and modifies the source field distribution to compensate for spatial non-uniformity introduced by optical components in the imaging system. In this study, we characterize the light coupling efficiency of the proposed design and compare the uniformity performance with the conventional method. EUCLID demonstrates a remarkable illumination improvement for multi-spectral imaging in both Nelsonian and Koehler alignment with a maximum spatial deviation of $\approx$ 1% across a wide field-of-view.

## Introduction

The illumination quality is of paramount importance for optical microscopy. Uneven illumination is a persistent issue for imaging techniques, particularly for widefield microscopy applications [1–3]. In many imaging applications, light sources, such as light emitting diodes (LEDs), lasers, or lamps, cannot be used directly since they provide uneven light output distribution. Thus, multiple optical elements are required to increase the quality of the illumination in typical imaging systems. However, these elements introduce a convoluted optical transfer function that affects the spatial and spectral uniformity. The most common visible effect of the non-ideal, limited numerical aperture (NA) optics is vignetting [1, 4]. For imaging applications, such as quantitative fluorescence microscopy, uniform excitation of the sample is critical for correct characterization of measured fluorophore response [5]. Another application example is optogenetics where spatiotemporal control of uniform and high-power excitation across the sample is crucial [6, 7]. Interferometric Reflectance Imaging Sensor (IRIS) [8], biosensing platform for accurate characterization of binding kinetics, provided the initial motivation for the illumination device described in this study. For the IRIS system, illumination uniformity drastically affects the detection accuracy and sensitivity for picometer level increments of

**Data Availability Statement:** All relevant data are within the paper and its Supporting information files.

**Funding:** M.S.Ü. received the following awards which funded the research work presented in this

study. National Institutes of Health, NIH B-BIC RADx (U54HL119145) (https://www.nih.gov/); National Science Foundation, NSF-TT PFI (1941195) (https://www.nsf.gov/). The funders had no role in study design, data collection and analysis, decision to publish, or preparation of the manuscript.

biomass accumulation. Therefore, design of the illumination source that can compensate for non-uniformities in the optical system is essential for numerous imaging and optical sensing applications.

Critical (or Nelsonian) alignment is one of the simplest illumination configurations where the light source is imaged on the object (sample) plane. Although this approach provides efficient light coupling in the system, which can be crucial for high signal-to-noise ratio, the underlying limitation is further compromising the light homogeneity. In modern light microscopy, Koehler illumination is preferred over critical illumination, where the image of the light source is defocused on the object plane and its conjugate planes. Therefore this method provides superior uniformity at the cost of total light power and requires additional optical elements. However, the common issue of vignetting persists in both configurations. To compensate for this effect, custom components have been introduced in literature. Mau et al. [5] proposed a laser scanning technique (ASTER), which employs 2D galvomirror to scan the entire field of view to have uniform excitation in single molecule localization microscopes. Coumans et al. [9] introduced two microlens arrays to a commercial epiflourescence microscope to flatten the illumination profile and Model et al [3] utilized concentrated fluorophore solutions to correct the spatial heterogeneity of the field, which cannot be applicable to the label-free microscopy. Thus, although the aforementioned methods increase illumination homogeneity, they require systems that significantly increase the overall complexity and cost. Computational methods have also been developed for image corrections [10] to avoid introducing optical complexity to the system, however they fail to fully compensate for the excitation field distortion [2] and real-time signal readings.

A particularly challenging microscopy method for illumination design is multi-spectral imaging (MSI), where field uniformity across the different spectral channels plays a key role for accurate interpretation of the signal [11]. Over the last 25 years, the advances in visible light emitting devices (LEDs) have presented unprecedented capabilities in multi-spectral light sources that provide compact, high-power and user-controlled color illumination. However, achieving homogeneous excitation for separate wavelength channels requires precise alignment for the illumination optics. The conventional multi-spectral illumination for MSI involves combining collimated beams at different wavelengths along a common optical path using dichroic beamsplitters and/or dichroic mirrors [12, 13]. These elements require diligent selection of illumination wavelengths, as well as increasing the total cost of the entire illumination system. The resulting illumination quality is also sensitive to optical alignment.

In this study, we present an LED based light source with an adjustable field profile, termed Effective Uniform Color-Light Integration Device (EUCLID). EUCLID, similar to a standard integrating sphere (IS), employs a hollow cavity with: a diffuse scattering surface, entrance ports for the light source and an exit port. The significant novelty of EUCLID is the introduction of conical geometry allowing for design optimization. The hollow cavity is engineered to improve the light coupling efficiency and field uniformity for different illumination configurations. We perform OpticStudio (Zemax) simulations to study the relationship between apex angle of the conical cavity of EUCLID and light output characteristics. We also examine and compare the field uniformity provided by different light sources under critical and Koehler illumination configurations. The primary finding of this study is the overall improvement in a critical illumination method with both single narrow band and multi-color imaging in comparison to the preferred standard in microscopy. We demonstrate the direct impact of EUCLID in imaging quality, providing a remarkable illumination uniformity with < 1% intensity deviation across different input source channels and ≈ 1% spatial light intensity variation within the full FOV (5mm x 7mm and 2mm x 2.8mm for 2x and 5x objective lenses, respectively) of our system. We finally test the color integration performance of EUCLID when it is

coupled to a multi-mode fiber to demonstrate its applicability for other illumination configurations.

## Materials and methods

### Design of EUCLID

Interferometric reflectance imaging sensor (IRIS) [8], introduces the fundamentals of utilizing diffuse reflections for mixing spatially separated LED sources and achieving a more uniform field profile. The system employs an integrating sphere in an unconventional manner. Integrating spheres evenly spread the input light by multiple reflections over the hollow cavity, therefore they are conventionally used for a variety of photometric or radiometric measurements. However, the IRIS systems have leveraged the beam produced by multiple diffusive reflection to obtain a light source with constant radiance (W/m$^2$/str) profile for all different color LED sources. This technique has also been evaluated for hyperspectral imaging by utilizing a diffuse scattering dome [11], and has shown as an effective method of achieving spatial homogeneity of discrete light sources. Therefore, we obtain effective mixing of multi-spectral LED sources in the conventional IRIS system by utilizing an integrating hollow cavity.

In IRIS systems, the sample is illuminated with common path reflectance mode, with either Koehler [14] or critical [8] illumination techniques. Although sufficient color mixing is achieved, the overall efficiency of light coupling is reduced given that the source profile (constant radiance) is not optimized for the finite-NA illumination optics. We have selected a conical hollow cavity geometry for EUCLID, to concentrate the light within the illumination path NA. The conical structure provides an intuitive geometry to confine the output rays into a prescribed cone. The geometry also enables a simple path for further modification of the cavity, since the light output emanates predominantly from the scattering on the base of the cone. The effect of conical geometry was studied by performing Non-Sequential ray tracing analysis on OpticStudio as it is described in *OpticStudio Simulations* section. Fig 1a shows the spherical

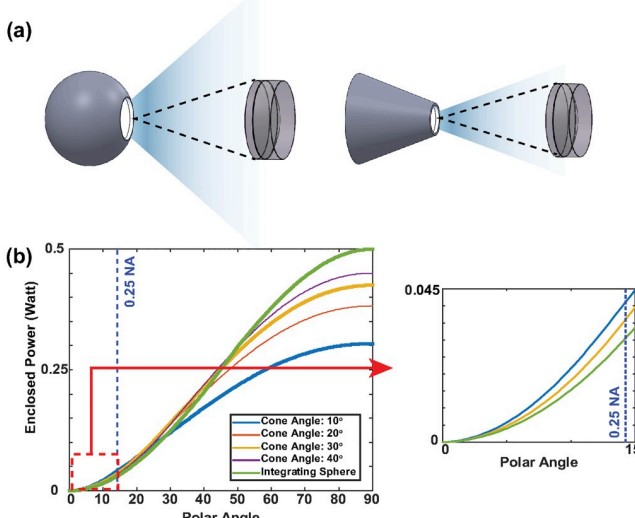

**Fig 1. Demonstration of light confinement offered by the conical geometry.** The sketch of simulated geometries (a). Total enclosed power with respect to polar angle graph obtained from OpticStudio simulations (b). Supplied power to the LIDs is set to 1 Watt.

and conical geometries and Fig 1b demonstrates the improvement of power confinement for a 0.25NA condenser lens. Conical LID provides 40% more power to the system than an ideal integrating sphere when the output port to internal area ratios are the same. However, for practical considerations, such as finite dimension of light input ports to match the LED size, external dimensions of the LIDs have limitations. For the same external sizes, a conical cavity has significantly smaller internal surface area compared to a spherical cavity, especially for small apex angles. For instance, the conical LIDs that we produced for this study have apex angles of 23° and 15° yielding ≈ 3 to ≈ 5 fold reduced in internal surface area, respectively. Considering total light output scales with the number of average reflections, LID efficiency depends on this port ratio by a power law [15]. In the *Power Efficiency due to Conical Geometry* section, we compared output efficiency of different geometric shapes with the same port ratio, drastically underestimating the improvement afforded by EUCLID design for practical LIDs.

After the light coupling analysis and validation of simulations for the conical cavity, we modify the geometry further, to shape the output field profile to achieve improved uniformity. In an aberration-free imaging system, irradiance of point $A$ on the image plane, $E_A$ is defined by

$$E_A = \int L(\theta, \phi) \, cos(\theta) \, d\Omega \tag{1}$$

where $L(\theta, \phi)$ is the radiance and the limit of this integral is determined by the physical limits of the exit pupil. If the radiance in Eq 1 is constant, i.e., Lambertian source is used, then irradiance at point $A$ is proportional to the projected solid angle subtended by the exit pupil according to point $A$ [16]. Thus, for an on-axis point, the exit pupil spans more area in the directional cosine space than for an off-axis point, which results in a bright spot at the center of the image plane. The irradiance uniformity degrades even further if other aberration causes, such as cosine-fourth law and pupil aberration, are considered [17]. To mitigate these issues, complex illumination source and lens designs are introduced and varying aperture configurations are studied in the literature [1, 9].

The simple design of EUCLID however, allows intuitive design optimization without further complicating the optics. Thanks to its conical geometry and small output port dimension, the radial distribution of the output depends only on the light rays scattered from the back surface (base of the cone). Thus, by engineering the back surface structure, the output radiance can be altered such that the finite-size aperture and aberration effects can be alleviated to achieve a uniform field profile. We opted for a design given in Fig 2 where the output radiance can be controlled and changed with a movable rod for different imaging systems with different exit pupil sizes. Note that the rod material is identical to the hollow cavity material.

The geometry of the cavity shape indicates that the majority of output light is scattered from the base of the cone. Therefore, the diameter of the movable rod, namely the diameter of the guiding hole on the back surface, given in Fig 2, affects the output radiance of EUCLID. It is possible to define the lower bound of output ray exit angles, $\theta_p$, when the rod position is at infinity, effectively creating a hole (namely a light trap) on the back surface. Using geometrical arguments (see S4 Fig), this pass angle, $\theta_p$, can be defined as

$$\theta_p = tan^{-1} \left( \frac{\frac{D}{2} - \frac{\varnothing_{output}}{2}}{h} \right) \tag{2}$$

where $D$ and $\varnothing_{output}$ are the diameters of guiding hole and output port, respectively and $h$ is the height of the conical cavity. In the presence of the movable rod, the output radiance can be

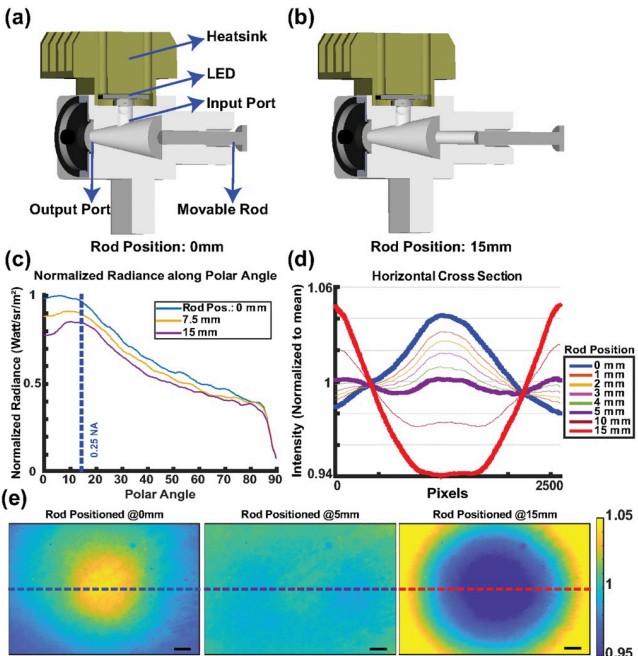

**Fig 2. Design and adjustable field performance of the EUCLID.** 3D model of EUCLID, adjustable rod positioned at 0mm (a) and 15mm (b). Simulation of output radiance profiles (c), the cross section (d) and heatmaps (e) of intensity images of the sample acquired from the setup given in S1 Fig for different rod positions. Rod positioned at 0mm corresponds to nominally flat base surface of the cone. The scale bar is 500$\mu$m.

tuned to compensate the vignetting effects. Specifically, EUCLID can compensate the drop in the high-frequency components of the illumination transfer function, typically due to finite-sized circular lenses employed in imaging systems and optical aberrations.

The selection of optimal $D$ value for two different illumination conditions is explained in the Supplementary Materials. We fabricated three different back surfaces with 3/16', 1/4" and 5/16" guiding holes which are compatible with commercially available PTFE rods and tested two conical LIDs with 15° and 23° apex angle combining them with the three back surfaces. This design flexibility allows EUCLID to perform robustly for illumination configurations composed of different optical elements (See S5 Fig).

## OpticStudio simulations

The effect of conical geometry was studied by performing Non-Sequential ray tracing analysis on OpticStudio for various light integrating devices (LIDs). For this purpose, we created 7 different conical light integrating devices and 1 integrating sphere whose port fractions, i.e. the ratio of output port and internal area, were designed to be identical. The apex angle of the conical LIDs was swept from 10° to 30° with a 5° step size and the output port diameter was selected as 5 mm for all devices. We used built-in objects to construct the LIDs and IS, and internal coatings were set to have a Lambertian scattering profile with 99% reflectivity. We placed two rectangular detectors on the back surface and output of the conical LIDs to validate light distributed evenly inside the conical structure. The total radiant flux within the far-field polar angles was calculated from the output light distribution, measured on the polar detector with a radius much larger than the output port dimensions ($r_{\text{polar detector}}$ = 60 mm). We have measured the output radiance of each LID from the rectangular detector placed on the output

ports. We also coupled EUCLID to a multi-mode fiber and achieved uniform output field profiles for different color channels. This indicates that EUCLID can be employed in applications where spatiotemporal control of uniform excitation is crucial, such as optogenetics. With temporal control of the different color LED input stimulation, EUCLID can also provide multi-color pulsed illumination and be employed in optogenetics applications since it offers high power illumination with even distribution on wide FOVs.

## Results and discussion

### Power efficiency due to conical geometry

We first analyzed the light confinement characteristic of conical geometry by OpticsStudio simulations. As the simulation parameters are explained in the *OpticStudio Simulations* section, we determined optimal apex angles for different commercial condenser lenses employed in 4-f Koehler system or critical illumination configuration. Given in Fig 3, the total coupled optical power is maximized when the apex angle of EUCLID matches with the acceptance angle of the first condenser lens.

To test and compare the experimental light coupling performances of EUCLID and IS, we have built a conventional IRIS system [8] with a 50mm condenser lens in the illumination path (See S1 Fig). In the imaging system, we employed a 2X 0.06NA (CFI Plan Achro) objective lens to focus the illumination light on the a reflective silicon substrate and image the sample plane with a CMOS camera (BFS-U3–70S7M-C). The CMOS readings were recorded for analyzing light coupling performances of LIDs under identical experimental settings. Table 1

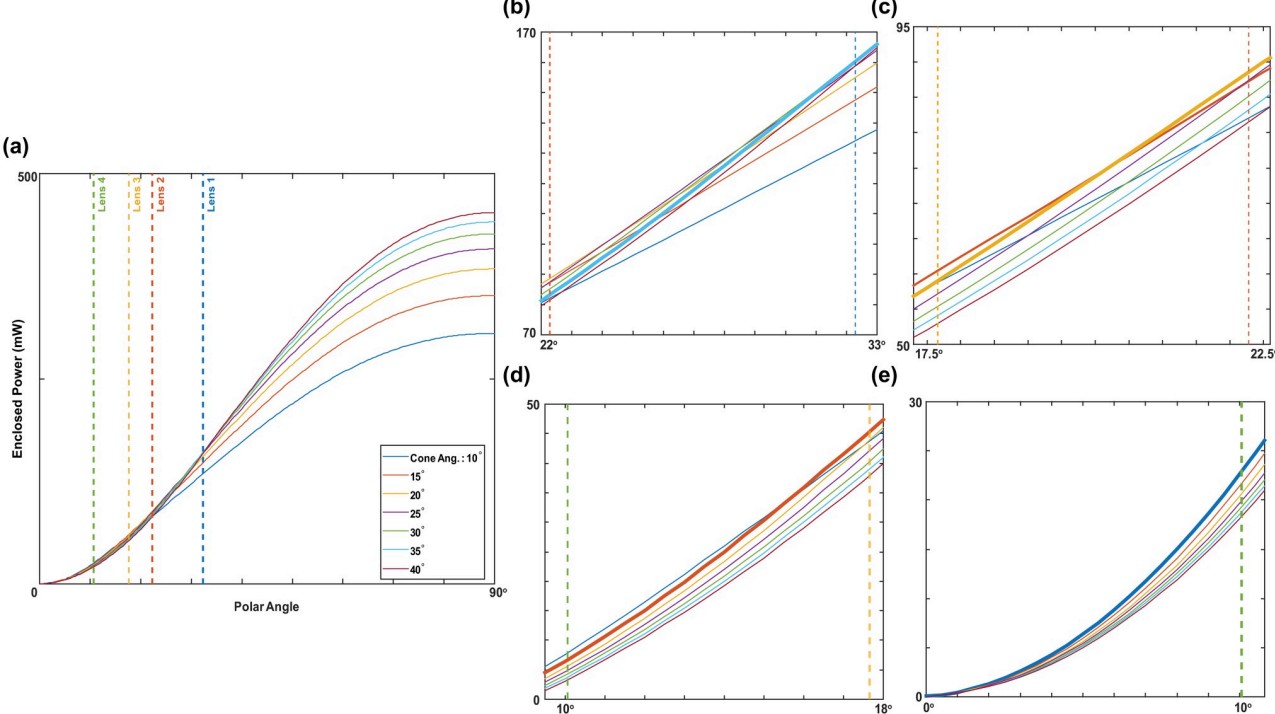

**Fig 3. Power coupling simulation results.** Total enclosed power with respect to polar angle graph calculated by Non-sequential ray tracing simulations. Full output characteristic of various conical LIDs (a). Zoomed sections of the left figure (b), (c), (d), (e). Best performing conical LID is indicated with the bold lines. The acceptance angles for the lenses are 32.29˚, 22.28˚, 17.66˚ and 10.70˚ for lens 1, 2, 3 and 4, respectively.

**Table 1. Light coupling performances of LIDs.**

| Device | Output Power (mW) | Power Ratio (EUCLID/IS) | Reading Ratio (EUCLID/IS) |
|---|---|---|---|
| IS | 102.4 | - | - |
| 23˚ Conic | 91.8 | 0.9 | 1.28 |
| 15˚ Conic | 76.5 | 0.75 | 1.58 |

summarizes the power analysis and compares the performances of an integrating sphere with two different conical LIDs. For this analysis, we fist measured the total output powers and compared the total intensity readings which were acquired from our setup. Although the total output flux from the conical LIDs are decreased, the confined flux within the system is increased by 28% and 58% for 23˚ conical and 15˚ conical LIDs, respectively. We fabricated our proof-of-concept LIDs from a PTFE block (reflectivity 95%) which cost under $50. Additionally, the output power of LIDs can be increased using commercial materials or coatings with higher reflectivity around 99% (i.e. Spectralon—LabSphere).

## Adjustable field profile

We have simulated the effect of the rod position on output radiance. As the results in Fig 2c (also S2 Fig) indicate, output radiance can be fully controlled by changing the rod position which is desired to compensate the cosine-fourth and vignetting effects for different imaging systems. Finally, to experimentally validate the performance of EUCLID, we have illuminated and imaged a flat $SiO_2$-Si substrate with the setup (see S1 Fig) and achieved an ultimate profile uniformity with 1.01% min-max deviation along the horizontal cross section (Fig 2d).

To test the uniformity performance of EUCLID, we have compared different illumination configurations for both critical and Koehler alignment. In this experimental setup (see S1 Fig) a 5x 0.15NA (TU Plan Flour) objective and a narrow band LED were used. All of the LIDs have identical output port diameter, resulting in identical illumination area. In order to quantify field uniformity for the resulting raw images, we have defined a parameter termed 'uniformity region', where the normalized intensity deviation is under a certain threshold (for instance 1% or 0.5%). The radii of these regions provide a quantitative metric to compare performance of different illumination configurations, the calculation for these regions is explained in the S1 Appendix. The uniformity regions for all sources are illustrated in Fig 4 and the calculated radii of 1% and 0.5% uniformity regions are given in Table 2. We have also compared the performance of EUCLID with the previously reported flat-fielding solutions. The results were given in S1 Table.

EUCLID, when the rod is positioned at 2mm and 6mm away from the back surface for Koehler and critical setup respectively, outperforms the integrating sphere and the standard

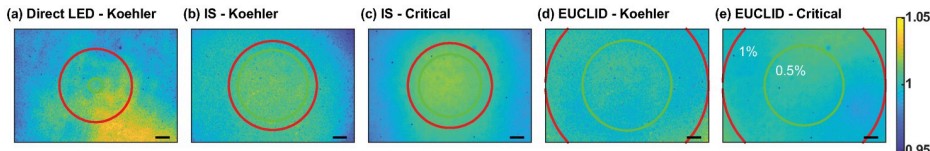

**Fig 4. Uniformity regions of different illumination sources.** The red and green circles indicate the area where the irradiance profile deviation is < 1% and < 0.5%, respectively. The uniform irradiance circles are shown for: direct LEDs (a), spherical LID (b), EUCLID (d) in Koehler alignment; spherical LID (c), EUCLID (e) in critical alignment. The scale bar is 200$\mu$m.

**Table 2. Radii of uniformity regions.**

| Source | Koehler | | Critical | |
|---|---|---|---|---|
| | **1%** | **0.5%** | **1%** | **0.5%** |
| **Direct LED** | 580 pxl | 110 pxl | - | - |
| **IS** | 705 pxl | 560 pxl | 670 pxl | 495 pxl |
| **EUCLID** | Full FOV (1581 pxl) | 710 pxl | Full FOV (1581 pxl) | 630 pxl |

approach of direct LEDs with a Koehler setup. Our design succeeded to obtain a 1.05% max-min deviation across the entire FOV.

## Multi-spectral illumination and fiber coupling

Uniform illumination profile for different wavelength channels is essential for MSI techniques. It is practically challenging to achieve a uniform multi-wavelength illumination profile with commercial flat-fielding solutions such as refractive or diffusive optics elements [18, 19] since one has to apply these solutions to different wavelength sources separately and then combine all the output profiles through dichroic elements on the same spatial location. Previously reported field correction systems, ASTER [5], Kohler integrator [20] or [21], can be used only if the different wavelength sources are combined through with such dichroic elements after their proposed field correction. Thus, this increases the cost of the total system significantly depending on the number of wavelengths required for a given setup. However, the light integrating devices allow mixing separate wavelength sources by multiple reflections inside the cavity and output only one uniform profile, which simplifies the correction method and reduces it to a single component. We have demonstrated the imaging improvement using an integrating cavity, with a 3-channel multi-spectral image cube we acquired from separate color channels. Using a monochrome camera with fixed optics, we have sequentially turned on different LED channels (633nm, 517nm and 453nm dominant, OSRAM LZ series) and acquired separate images. We then created the pseudo RGB images after normalizing each channel to account for variations in die brightness. The pseudo RGB images were created for three different illumination path conditions: direct LEDs, spherical LID, EUCLID in Koehler alignment. To create these images, we have normalized each monochrome measurement with respect to the peak value of their histograms, i.e. their means. Then, we have analyzed the MSI uniformity performances by examining the brightness deviations with respect to other channels across the FOV and the standard deviation along the color channel dimension was recorded for each pixel. Fig 5 shows the horizontal cross sectional view of color deviation for each condition. As expected, without the use of an integrating device, we observed a poor color mixing performance and effectively a reduced uniformly illuminated FOV. EUCLID showed a remarkable color uniformity performance with an average deviation of less than 1%.

Multi-mode fibers can be used for compact and convenient multi-spectral light delivery in microscopy [22]. We studied the spectral mixing performance of EUCLID for fiber-optic applications, as an additional evaluation. We coupled the light sources to the fiber facet (Thorlabs M93L02, $\varnothing_{core}$ = 1.5mm, 0.39NA), and imaged the output of the fiber to analyze field profiles with respect to separate source channels (See S4 Fig). We used a lens pair, 200mm and 30mm, to couple the light sources into the core, by matching the fiber NA. The system was aligned for the 453 nm dominant channel for both illumination systems and the optical alignment was kept fixed. We then acquired images for the remaining spectral channels and created the psuedo RGB images shown in Fig 6. The direct LED coupling suffered from spatial

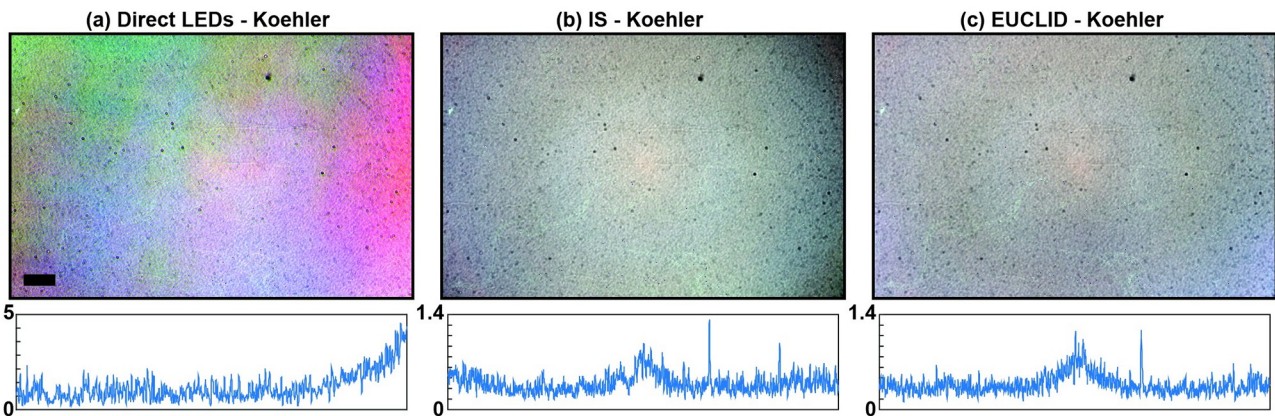

**Fig 5.** Pseudo RGB images (top) and respective spectral deviation %, across the horizontal cross section (bottom). Direct LEDs (a), Spherical LID (b), EUCLID (c) in Koehler alignment. Contrasts are enhanced for better visualisation. The scale bar is 200$\mu$m.

separation of LED dies, resulting in an uneven field with peripheral and central concentration of separate channels. EUCLID maintained a uniform spatial and spectral profile, like an IS. However, EUCLID can confine more light into smaller NA which would increase the coupling efficiency. It is also possible to connect fibers directly to the output of EUCLID when its apex angle and output diameter is designed to match with fiber dimensions and its NA.

## Conclusion

In summary, we have demonstrated the light coupling and multi-spectral illumination performance of EUCLID. Although certain parameters of EUCLID has to be chosen in the fabrication process, the cost-effective material and modular design allows for testing different combinations of these parameters to effectively optimize for a given system. We have validated the feasibility of EUCLID as the light source in various widefield imaging systems, and an effective spatial uniformity correction. Moreover, its design also allows to combine different conical parts with various back surface structures. This intuitive conical geometry and an adjustable cavity serves the purpose of introducing a novel design parameter for light illumination devices and EUCLID can accommodate and correct for imaging systems with different effective transfer functions, elements and alignment. With temporal control of the different color LED input stimulation, EUCLID can provide multi-color pulsed illumination over a wide FOV.

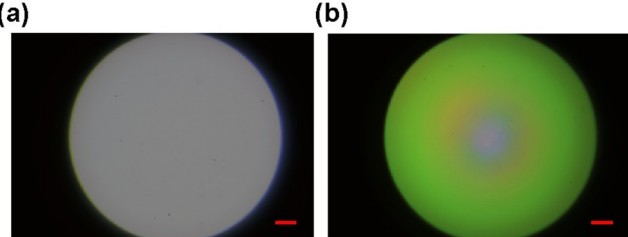

**Fig 6. Pseudo RGB images of fiber facet.** Output of EUCLID (a) and direct LEDs (b) are coupled to the input end of the fiber. The scale bar is 200$\mu$m.

## Supporting information

**S1 Fig. Optical setup.** Detailed schematics of the imaging setups. Left: with critical illumination, Right: Koehler illumination. The light integration devices (LIDs) or direct LED dies were aligned to the imaging optics in identical conditions to acquire field profiles.
(TIF)

**S2 Fig. Output radiance simulations.** Output Radiance cross sections of EUCLID with 5 mm output port and rod diameter when rod is positioned different locations. Left: Output radiance for all polar angles. Acceptance angle is defined by lens 3 in Fig 3. Right: Zoomed section of left graph for angles that lies within the acceptance angle.
(TIF)

**S3 Fig. Fiber performance setup.** Schematic of the imaging setups for fiber alignment, with direct LEDs (right) and with EUCLID (left). The light output of direct LED dies and the EUCLID were coupled to the fiber tip by a lens pair for demagnification. The output of the fiber tip is then imaged by using the same collection optics as previous setups.
(TIF)

**S4 Fig. Ray approach for EUCLID.** Toy picture of the EUCLID geometry where the output ray with the minimum exit angle is indicated. $h$ is the height, $\varnothing_{output}$ is the output port and $D$ is the guiding hole diameter of the EUCLID.
(TIF)

**S5 Fig. Performance of non-optimized EUCLID.** Horizontal cross sections of two different non-optimized EUCLID in different illumination configurations. EUCLID with rod diameter 1/4" (a,c) is tested under critical illumination configuration. Normalized heatmap (c) and corresponding horizontal cross sections for different rod positions (a) are indicated. EUCLID with rod diameter 3/16" (b,d) is tested under Koehler illumination configuration. Normalized heatmap (d) and corresponding horizontal cross sections for different rod positions (b) are indicated.
(TIF)

**S1 Table. Comparison of different flat-fielding systems.** First two metrics in the table, plateau uniformity and flatness factor, is defined in the ISO 13694:2000 standard [23] for continuous waves. The metric values, expect EUCLID's and IS's, is taken from corresponding references [20, 21, 24]. The ISO values for [18–20] is characterized without the effect of the external optical element, such as objective and tube lenses, dichorics or beam splitters, etc. The commercial refractive elements, TopShape and PiShaper, also require spatial filtering of the input beam. The values for the remaining are calculated in a common epi-illumination system, where the effects of the external components are considered. Performance metrics of different flat-top illumination system, where the effects of the external components are considered.
(PDF)

**S1 Appendix. Custom MATLAB Code to Determine Uniformity Regions.** To quantify the illumination profile uniformity, we have defined uniformity regions as the largest area enclosed, that yield a normalized intensity deviation under 1% or 0.5%. We calculated the radii of the regions with a custom MATLAB algorithm. The algorithm initializes the ROI with a circular region (radius of 25 pixels) and compares the mean values of the current ROI to the annulus (with a thickness of 10 pixels) just outside of the initial region. If the average absolute brightness deviation of the annulus, compared to the previous ROI, is within the specified range (1 or 0.5%) the algorithm continues to evaluate the absolute brightness deviation in the

next annulus with the same thickness, just outside of the previous annular ROI. The algorithm iterates until the specified annular ring indicates that the uniformity level has decreased below the specified range or the radii hits the corner of the FOV. The uniformity regions for all sources are illustrated in Fig 4 and the calculated radii of 1% and 0.5% uniformity regions are given in Table 2 in the manuscript. We also applied Gaussian filter ($\sigma = 5$) to remove the small artefacts on the sample, such as dust particles, etc.
(PDF)

**S2 Appendix. Geometric optics approach to determine optimal EUCLID parameters.**
(PDF)

## Acknowledgments

The authors acknowledge Nevzat Yaraş of iRiS Kinetics for valuable help in the fabrication of LIDs.

## Author Contributions

**Conceptualization:** İris Çelebi, Mete Aslan, M. Selim Ünlü.

**Data curation:** İris Çelebi, Mete Aslan.

**Formal analysis:** Mete Aslan.

**Funding acquisition:** M. Selim Ünlü.

**Investigation:** İris Çelebi.

**Resources:** M. Selim Ünlü.

**Supervision:** M. Selim Ünlü.

**Validation:** İris Çelebi, Mete Aslan.

**Visualization:** Mete Aslan.

**Writing – original draft:** İris Çelebi, Mete Aslan.

**Writing – review & editing:** İris Çelebi, Mete Aslan, M. Selim Ünlü.

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
