## [Decision Letter · Decision Letter 0]

16 Jan 2023

PONE-D-22-33508A spatially uniform illumination source for widefield multi-spectral optical microscopyPLOS ONE

Dear Dr. Celebi,

Thank you for submitting your manuscript to PLOS ONE. After careful consideration, we feel that it has merit but does not fully meet PLOS ONE’s publication criteria as it currently stands. Therefore, we invite you to submit a revised version of the manuscript that addresses the points raised during the review process.

We look forward to receiving your revised manuscript.

Kind regards,

Kun Chen, Ph.D

Academic Editor

PLOS ONE

Journal Requirements:

3. Thank you for stating the following in the Competing Interest section: 

"M.S.U. is the founder and CEO at iRiS Kinetics. All

authors declare no competing interests."

We note that one or more of the authors are employed by a commercial company: iRiS Kinetics

(2) Please also provide an updated Competing Interests Statement declaring this commercial affiliation along with any other relevant declarations relating to employment, consultancy, patents, products in development, or marketed products, etc.  

Within your Competing Interests Statement, please confirm that this commercial affiliation does not alter your adherence to all PLOS ONE policies on sharing data and materials by including the following statement: ""This does not alter our adherence to  PLOS ONE policies on sharing data and materials.” (as detailed online in our guide for authors http://journals.plos.org/plosone/s/competing-interests) . If this adherence statement is not accurate and  there are restrictions on sharing of data and/or materials, please state these. Please note that we cannot proceed with consideration of your article until this information has been declared.

Reviewers' comments:

Reviewer's Responses to Questions

**Comments to the Author**

1. Is the manuscript technically sound, and do the data support the conclusions?

Reviewer #1: Yes

Reviewer #2: Partly

2. Has the statistical analysis been performed appropriately and rigorously? 

Reviewer #1: Yes

Reviewer #2: Yes

3. Have the authors made all data underlying the findings in their manuscript fully available?

Reviewer #1: Yes

Reviewer #2: Yes

4. Is the manuscript presented in an intelligible fashion and written in standard English?

Reviewer #1: Yes

Reviewer #2: Yes

5. Review Comments to the Author

Reviewer #1: Celebi et al present a novel method to generate a uniform illumination profile from an LED for high-quality wide field experiments. EUCLID, their method, uses a fixed optical element to create this uniformity which contrasts with existing methods that either use fixed, but expensive, beam shapers, or movable elements, such as a vibrating multimode fibre. The manuscript is well written, but the following comments need to be considered before I can recommend publication:

1, the value proposition of the presented method needs to be rethought and well articulated. There are existing methods (e.g., Holo Or beam shaper) that is a fixed, diffusive elements that produces a high quality top hat profile from a Gaussian input. How is this different - is it the price, the fact that it can be used across a range of wavelength simultaneously, the fact it can be used with an LED, etc.? I suggest the authors to include a table that summarises published/commercial methods and compares the advantages/disadvantages of each with EUCLID.

2, the authors do not include a full list of references to methods developed for uniform illumination in wide field microscopy. There is FIFI from the lab of Suliana Manley at EPFL, efficient homogenous illumination from the lab of Jonas Ries and an economic, square-shaped illumination from the lab of David Klenerman - and more. Authors would need to list those reference and to include them in the table above.

3, LEDs are not used for single molecule microscopy as much as lasers due to their broad spectral properties and complexity in focusing/collimating them for high-density illumination (in addition to several other reasons). How does EUCLID address the problem of non-uniform illumination in single molecule microscopy?

4, the majority of the plots are incredibly hard to read (possibly due to PDF rendering as well as to small plot text, etc.). Also some plots are not labelled A,B, C, etc… (e.g.,

Reviewer #2: In the manuscript "A spatially uniform illumination source for widefield multi-spectral optical microscopy", the authors present a novel illumination design for flat illumination in widefield and multi-spectral microscopy applications. The main improvement comes from a neat improvement in the geometry of light integrating device, allowing it to better match the rest of the optical elements in the microscope, mainly the numerical aperture of the condenser lens. The performance of this device, termed EUCLID, is demonstrated via simulations and integration into a real imaging microscope. The work provides a novel and modular approach for flattening uneven LED illumination from multiple sources.

Major comments:

- While EUCLID is a novel and intelligent approach towards flat multi-spectral illumination, it is unclear how previous flat-fielding solutions fail in this mission, or how the performance of EUCLID is superior to previous developments. The authors mention other flat-fielding solutions such as the citations [3-7] in the introduction (lines 18-33) but do not comment on why these are insufficient or inferior to EUCLID in providing flat illumination and in the context of multi-spectral applications. Intuitively, I do not see why these solutions (and some others such as flat-fielding optics, some summarised in Khalid A. Ibrahim, Dora Mahecic, and Suliana Manley, "Characterization of flat-fielding systems for quantitative microscopy," Opt. Express 28, 22036-22048 (2020)) would not perform well in the context of multi-spectral imaging. This is important for fully appreciating the novelty and necessity of the presented work.

- Another aspect which was better covered in the above mentioned works is the compatibility of many light sources with laser and coherent illumination. It is unclear whether EUCLID would perform as well in these contexts. The authors should better comment on potential limitations of EUCLID.

- A better explanation of the optimal adjustment of the pass angle for Koehler alignment in lines 124-126 is needed. Also, a clarification on how D is chosen.

- Also, is there any explanation for why b) appears to perform better than e) in Figure 5?

- How does an integrating sphere perform when coupled with the multimode fiber (Figure 6)?

Minor comments:

- In the introduction (lines 4-7), the authors attribute the uneven illumination to the imperfect optical transfer functions of the optical elements between the light source, sample and camera. However, this neglects an important contribution of the light source itself - many light sources including LEDs, lamps and lasers have uneven illumination profiles which significantly contribute to the uneven excitation, arguably more so than the optical elements themselves. This should be clarified in the introduction.

- Similarly, the authors state that the difficulty in "achieving homogeneous excitation for separate wavelength channels" is "simply due to spatial separation of the source elements" (lines 39-41). Again, I disagree, but instead think the profile of the excitation sources to be much more problematic in achieving flat illumination.

- Throughout the manuscript, many different units are used in plots to compare the performance of EUCLID to the integrating sphere and between designs. The units vary a lot (radiant flux, radiance, intensity) - which at times makes it hard to understand how much more efficient EUCLID is. For instance, in Fig 1, how much more efficient is EUCLID compared to the integrating sphere is not obvious. It would be advantageous to include more intuitive measurements too in the text, such as a % improvement over other methods.

- Despite the movable rod, the design of EUCID (angle, height, diameters of the guiding hole and output port) are rigid and have to be chosen during the manufacturing process. I imagine this is a significant limitation worth mentioning. Are there any potential solutions for making some of these parameters more flexible?

- IS is mentioned in line 141 without any prior definition. I'm assuming it refers to the integrating sphere - define.

- The order of appearance of references is not uniformly increasing and makes the cross-referencing with the bibliography more difficult.

- Clarify in Figure 2 whether the data in c-e is real or simulated data.

- Text in Figure 3 is difficult to see.

- Panels in Figure 5 skip the letter d.

- All figure are references as Fig. X except for Figure 1 which is mentioned in full.

6. PLOS authors have the option to publish the peer review history of their article (what does this mean?). If published, this will include your full peer review and any attached files.

Reviewer #1: No

Reviewer #2: No

---

## [Author Response · Author response to Decision Letter 0]

20 Apr 2023

Reviewer #1: Celebi et al present a novel method to generate a uniform illumination profile from an LED for high-quality wide field experiments. EUCLID, their method, uses a fixed optical element to create this uniformity which contrasts with existing methods that either use fixed, but expensive, beam shapers, or movable elements, such as a vibrating multimode fibre. The manuscript is well written, but the following comments need to be considered before I can recommend publication:

1. The value proposition of the presented method needs to be rethought and well-articulated. There are existing methods (e.g., Holo Or beam shaper) that is a fixed, diffusive elements that produces a high quality top hat profile from a Gaussian input. How is this different - is it the price, the fact that it can be used across a range of wavelength simultaneously, the fact it can be used with an LED, etc.? I suggest the authors to include a table that summarizes published/commercial methods and compares the advantages/disadvantages of each with EUCLID.

Answer:

As the reviewer suggests, EUCLID can operate over a broad spectrum which is defined by the reflectance and properties of the PTFE material (>%95 for 200 - 2500 nm). In contrast, diffractive or refractive optical element solutions can produce uniform illumination only at their design wavelength for a Gaussian beam. These elements might also require precise alignment and spatial filtering to the input beam, which would increase the cost of the entire illumination system. 

EUCLID system is shown to perform with multi-color LED sources demonstrating its ability to homogenize light output from spatially separated sources. 

We also thank reviewer for suggesting to create a table and for the additional references. We included a comparison table in the Supplementary Information section.

2. The authors do not include a full list of references to methods developed for uniform illumination in wide field microscopy. There is FIFI from the lab of Suliana Manley at EPFL, efficient homogenous illumination from the lab of Jonas Ries and an economic, square-shaped illumination from the lab of David Klenerman - and more. Authors would need to list those reference and to include them in the table above.

Answer:

We thank the reviewer for these additional references. We created a comparison table and included it in the supplementary materials to discuss different advantages and metrics.

3. LEDs are not used for single molecule microscopy as much as lasers due to their broad spectral properties and complexity in focusing/collimating them for high-density illumination (in addition to several other reasons). How does EUCLID address the problem of non-uniform illumination in single molecule microscopy?

Answer:

We appreciate the reviewer's concern about using EUCLID in SMLM. Laser illumination and SMLM applications are outside of the current scope of our study. EUCLID is developed to improve sensitivity in the IRIS system, where LEDs are preferred over lasers since they provide uniform, high-power illumination with sufficiently narrow spectrum. We also noted that LEDs are also preferred in optogenetics applications due to their overall cost and high modulation capabilities [ref 6-7 of the main manuscript]. Regarding single molecule and super-resolution microscopy, in our understanding, the critical issue is uniform illumination in a narrow spectral range. Temporal coherence provided by lasers is not a concern. EUCLID provides very good uniformity and allows for using high-power LED sources (~10W). Even with a spectral filter, LED-EUCLID combination could be a low-cost alternative for single-molecule microscopy providing superior uniformity in wide-field imaging. 

4. The majority of the plots are incredibly hard to read (possibly due to PDF rendering as well as to small plot text, etc.). Also some plots are not labelled A,B, C, etc… (e.g.,

Answer:

We thank the reviewer for noting that. We fixed the labeling issue and converted the figures to .eps files. 

Reviewer #2: In the manuscript "A spatially uniform illumination source for widefield multi-spectral optical microscopy", the authors present a novel illumination design for flat illumination in widefield and multi-spectral microscopy applications. The main improvement comes from a neat improvement in the geometry of light integrating device, allowing it to better match the rest of the optical elements in the microscope, mainly the numerical aperture of the condenser lens. The performance of this device, termed EUCLID, is demonstrated via simulations and integration into a real imaging microscope. The work provides a novel and modular approach for flattening uneven LED illumination from multiple sources.

Major comments:

- While EUCLID is a novel and intelligent approach towards flat multi-spectral illumination, it is unclear how previous flat-fielding solutions fail in this mission, or how the performance of EUCLID is superior to previous developments. The authors mention other flat-fielding solutions such as the citations [3-7] in the introduction (lines 18-33) but do not comment on why these are insufficient or inferior to EUCLID in providing flat illumination and in the context of multi-spectral applications. Intuitively, I do not see why these solutions (and some others such as flat-fielding optics, some summarised in Khalid A. Ibrahim, Dora Mahecic, and Suliana Manley, "Characterization of flat-fielding systems for quantitative microscopy," Opt. Express 28, 22036-22048 (2020)) would not perform well in the context of multi-spectral imaging. This is important for fully appreciating the novelty and necessity of the presented work.

Answer:

 We appreciate these concerns. As we provided in S1 Table, EUCLID provides approximately 6 times more uniform illumination profile than the Kohler Integrator, developed by Manley’s group, and 2 times better performance than the double MLA system, developed by Coumans et al. We also provided in Figure 5 of the main manuscript that EUCLID performs robustly at the different wavelengths across the visible spectrum.

 We also agree with the reviewer's comment about the difficulties of providing multi-spectral illumination with the mentioned method. Our technique provides a solution for multi-wavelength imaging across a broad range of wavelengths. EUCLID offers a practical approach where multi-spectral sources can be combined inside the EUCLID through multiple random reflections and output from the same port. This will decrease the total cost of such systems and facilitate the alignment procedure. We revised the manuscript accordingly.

- Another aspect which was better covered in the above mentioned works is the compatibility of many light sources with laser and coherent illumination. It is unclear whether EUCLID would perform as well in these contexts. The authors should better comment on potential limitations of EUCLID.

Answer:

 The scope of this study is motivated by IRIS and optogenetics applications of EUCLID, where temporal coherent illumination is not required. LEDs are employed in those applications since they can provide sufficiently high optical power and high modulation rate in a cost effective manner. 

We appreciate this concern of the reviewer. We understand that the critical issue in PALM/STORM based super-resolution microscopy is uniform illumination in a narrow spectral range. Temporal coherence provided by lasers is not required. EUCLID provides very good uniformity and allows for using high-power LED sources (~10W). Even with a spectral filter, LED-EUCLID combination could be a low-cost alternative for single-molecule microscopy providing superior uniformity in wide-field imaging.

- A better explanation of the optimal adjustment of the pass angle for Koehler alignment in lines 124-126 is needed. Also, a clarification on how D is chosen.

Answer:

 We thank the reviewer for this comment and adjusted the lines [147-157 in the main manuscript] accordingly. We also include a supplementary section about how D affects the output profile in Koehler configuration.

- Also, is there any explanation for why b) appears to perform better than e) in Figure 5? 

Answer:

 We thank the reviewer for noting this mistake. The previous version of the figure compares illumination performance of sources utilizing different mirror chips. We noted that the same location of the same chip has to be compared under a given illumination condition to make a quantitative comparison between color uniformity profiles of different illumination devices. So, we updated Figure 5 accordingly. We also added a brief explanation about how we created the pseudo-RGB images. The figure proves that EUCLID mixes the input light uniformly through multiple random reflections like an integrating sphere and can be employed as an light integrating device.

- How does an integrating sphere perform when coupled with the multimode fiber (Figure 6)?

Answer:

 We clarify this in the main manuscript. Please see lines between [262-265] in the main manuscript.

Minor comments:

- In the introduction (lines 4-7), the authors attribute the uneven illumination to the imperfect optical transfer functions of the optical elements between the light source, sample and camera. However, this neglects an important contribution of the light source itself - many light sources including LEDs, lamps and lasers have uneven illumination profiles which significantly contribute to the uneven excitation, arguably more so than the optical elements themselves. This should be clarified in the introduction.

Answer:

 We thank the reviewer for this comment. We included these comments in the introduction (lines 4-10).

‘In many imaging applications, light sources, such as light emitting diodes (LEDs), lasers, or lamps, cannot be used directly since they provide uneven light output distribution. Thus, multiple optical elements are required to increase the quality of the illumination in typical imaging systems. However, these elements introduce a convoluted optical transfer function that affects the spatial and spectral uniformity.’

- Similarly, the authors state that the difficulty in "achieving homogeneous excitation for separate wavelength channels" is "simply due to spatial separation of the source elements" (lines 39-41). Again, I disagree, but instead think the profile of the excitation sources to be much more problematic in achieving flat illumination.

Answer:

 We thank the reviewer for pointing out this misunderstanding. With the quoted statement, we want to indicate why the output profiles of excitation sources are not uniform when comparing separate channels. For instance, multi-color LED sources are comprised of color dyes that are located on the different spatial locations. This yields an uneven output distribution of separate wavelength channels when the optical alignment is not readjusted for the operating color die. Specifically, the uniformity in this claim refers to the deviation of field profile with respect to the different color channels. We fixed these claims in the introduction and focused on alignment problems in the Introduction and Multi-spectral Illumination and Fiber Coupling sections.

- Throughout the manuscript, many different units are used in plots to compare the performance of EUCLID to the integrating sphere and between designs. The units vary a lot (radiant flux, radiance, intensity) - which at times makes it hard to understand how much more efficient EUCLID is. For instance, in Fig 1, how much more efficient is EUCLID compared to the integrating sphere is not obvious. It would be advantageous to include more intuitive measurements too in the text, such as a % improvement over other methods.

Answer:

 We thank reviewers for pointing out this issue. We included quantitative comparison in the main manuscript (lines 99-109).

- Despite the movable rod, the design of EUCID (angle, height, diameters of the guiding hole and output port) are rigid and have to be chosen during the manufacturing process. I imagine this is a significant limitation worth mentioning. Are there any potential solutions for making some of these parameters more flexible?

Answer:

 We clarify this in the Conclusion section. EUCLID performs sufficiently even if it is not specifically optimized for a given system. In other words, the design tolerances are forgiving if the device is not fabricated with the optimal values. 

We included the uniformity performance of non-optimized EUCLID in S5 Fig.

- IS is mentioned in line 141 without any prior definition. I'm assuming it refers to the integrating sphere - define.

Answer:

 We thank the reviewer for pointing this out.

- The order of appearance of references is not uniformly increasing and makes the cross-referencing with the bibliography more difficult.

Answer:

 We thank the reviewer for pointing that out. We fixed this mistake.

- Clarify in Figure 2 whether the data in c-e is real or simulated data.

Answer:

 We adjusted the figure caption accordingly.

- Text in Figure 3 is difficult to see.

Answer:

 We included an .eps file for better rendering and increased the font size.

- Panels in Figure 5 skip the letter d.

Answer:

 We have corrected labeling in figure 5.

- All figure are references as Fig. X except for Figure 1 which is mentioned in full.

Answer:

 We thank the reviewer for pointing this out. We used ‘Fig.’ instead of ‘Figure’ to refer to a figure in the text which is stated in the PLOS one guidelines.

---

## [Decision Letter · Decision Letter 1]

30 May 2023

A spatially uniform illumination source for widefield multi-spectral optical microscopy

PONE-D-22-33508R1

Dear Dr. Celebi,

We’re pleased to inform you that your manuscript has been judged scientifically suitable for publication and will be formally accepted for publication once it meets all outstanding technical requirements.

Kind regards,

Kun Chen, Ph.D

Academic Editor

PLOS ONE

Additional Editor Comments (optional):

Reviewers' comments:

Reviewer's Responses to Questions

**Comments to the Author**

1. If the authors have adequately addressed your comments raised in a previous round of review and you feel that this manuscript is now acceptable for publication, you may indicate that here to bypass the “Comments to the Author” section, enter your conflict of interest statement in the “Confidential to Editor” section, and submit your "Accept" recommendation.

Reviewer #2: All comments have been addressed

2. Is the manuscript technically sound, and do the data support the conclusions?

Reviewer #2: Yes

3. Has the statistical analysis been performed appropriately and rigorously? 

Reviewer #2: Yes

4. Have the authors made all data underlying the findings in their manuscript fully available?

Reviewer #2: Yes

5. Is the manuscript presented in an intelligible fashion and written in standard English?

Reviewer #2: Yes

6. Review Comments to the Author

Reviewer #2: The Competing interest statement suggests some competing interest since in the comments to the reviewers, the work is partly motivated by:

"EUCLID is developed to improve sensitivity in the IRIS system, ..."

and

"The scope of this study is motivated by IRIS and optogenetics applications of EUCLID..."

As M.S.U. has a listed competing interest, it might be worth rewording "All authors declare no competing interests." to something more like "All other authors declare no competing interests."

7. PLOS authors have the option to publish the peer review history of their article (what does this mean?). If published, this will include your full peer review and any attached files.

Reviewer #2: No

---

## [Editor Report · Acceptance letter]

4 Jul 2023

PONE-D-22-33508R1 

A spatially uniform illumination source for widefield multi-spectral optical microscopy 

Dear Dr. Celebi:

I'm pleased to inform you that your manuscript has been deemed suitable for publication in PLOS ONE. Congratulations! Your manuscript is now with our production department. 

Kind regards, 

on behalf of

Dr. Kun Chen 

Academic Editor

PLOS ONE